# P300 Event-Related Potential Predicts Cognitive Dysfunction in Patients with Vestibular Disorders

**DOI:** 10.3390/biomedicines11092365

**Published:** 2023-08-24

**Authors:** Xiaobao Ma, Jiali Shen, Jin Sun, Lu Wang, Wei Wang, Kuan He, Xiangping Chen, Qin Zhang, Yulian Jin, Dekun Gao, Maoli Duan, Jun Yang, Jianyong Chen, Jingchun He

**Affiliations:** 1Department of Otorhinolaryngology-Head and Neck Surgery, Xinhua Hospital, Shanghai Jiaotong University School of Medicine, Shanghai 200092, China; maxiaobao@xinhuamed.com.cn (X.M.); shenjiali@xinhuamed.com.cn (J.S.); sunjin7018@163.com (J.S.); wlu6982@163.com (L.W.); wangwei04@xinhuamed.com.cn (W.W.); he241368@163.com (K.H.); xhcxp@163.com (X.C.); zhangqin5235@126.com (Q.Z.); jinyulian8548@xinhuamed.com.cn (Y.J.); gaodekun1994113@163.com (D.G.); yangjun@xinhuamed.com.cn (J.Y.); 2Ear Institute, Shanghai Jiaotong University School of Medicine, Shanghai 200092, China; 3Shanghai Key Laboratory of Translational Medicine on Ear and Nose Diseases, Shanghai 200092, China; 4Ear Nose and Throat Patient Area, Trauma and Reparative Medicine Theme, Karolinska University Hospital, 171 76 Stockholm, Sweden; maoli.duan@ki.se; 5Division of Ear, Nose and Throat Diseases, Department of Clinical Science, Intervention and Technology, Karolinska Institutet, 171 77 Stockholm, Sweden

**Keywords:** cognitive impairment, vertigo and imbalance, P300, MMSE, older adults

## Abstract

Objective: Our aim was to determine the correlation between cognitive impairment and P300 event-related potential (ERP) in older adults with vertigo and imbalance, which further provides a reference for clinical diagnosis and patients’ rehabilitation. Methods: A total of 79 older adult patients with vertigo and imbalance in our outpatient department from January 2022 to December 2022 were selected and divided into the mild group (n = 20), moderate group (n = 39), and severe group (n = 20) according to the Dizziness Handicap Inventory (DHI). The auditory P300 component of event-related potentials (ERPs), Generalized Anxiety Disorder Questionnaire-7 (GAD-7), Patient Health Questionnaire-9 (PHQ-9), and Mini-Mental State Examination (MMSE) were used to evaluate depression, anxiety, and cognitive function in these patients, respectively. Results: The P300 latencies of the different severity groups were 292 ± 10 ms, 301 ± 8 ms, and 328 ± 5 ms, respectively, and the differences were statistically significant (*p* = 0.010). The P300 amplitudes of the different severity groups were 14.4 ± 2.6 μV, 3.9 ± 0.8 μV, and 5.1 ± 1.4 μV, respectively, and the differences were also statistically significant (*p* = 0.004). There was no statistically significant difference in the DHI evaluation or VAS visual simulation scoring between the two groups (*p* = 0.625, and 0.878, respectively). Compared with the short-course group, the long-course group showed prolonged P300 latency and decreased amplitude, higher scores in PHQ-9 and GAD-7, and lower scores in MMSE, and all the differences were statistically significant (*p* = 0.013, 0.021, 0.006, 0.004, and 0.018, respectively). Conclusion: Older patients with more severe symptoms of vertigo and imbalance are at higher risk of developing abnormal cognitive function. The P300 can be used as an objective neurophysiological test for the assessment of cognitive function relevant to elderly patients with vertigo and imbalance.

## 1. Introduction

Dizziness, vertigo, and balance disorders related to the vestibular system are very common among the elderly, with a lifetime prevalence of up to 25% [1]. In the United States, 19.6% of people aged 65 and older have experienced dizziness, vertigo, or balance problems in the past year, including difficulty with unsteadiness (68%), walking on uneven surfaces (55%), vertigo (30%), and faintness (30%) [2]. Furthermore, the prevalence of these symptoms increases rapidly with age. Research shows that the incidence of dizziness and vertigo among people aged 65 and older is 20–30%, while for those over 80 years old, it exceeds 50% [3]. Previous studies have shown that vestibular-related dizziness and vertigo can cause physical, emotional, and functional impairments in patients, severely affecting work efficiency. In one study, 51% of patients with dizziness and vertigo were unable to work normally or experienced a significant decline in work efficiency, and 12% showed complete disability [4]. Dizziness and vertigo can worsen the quality of life of the elderly and increase the risk of falls [5].

In addition, dizziness and vertigo are also associated with cognitive dysfunction [6]. An increasing amount of research suggests that the vestibular system plays an important role in cognition [7]. Behavioral studies on animals and humans with vestibular pathology indicate deficits in spatial orientation and memory [8]. Furthermore, new evidence suggests that the loss of vestibular function may lead to long-term cognitive impairments, not only limited to visual–spatial ability but also including other cognitive domains such as attention, memory, and executive function [9]. When vestibular damage occurs, the anatomical structure and functional connectivity of the brain cortical regions receiving vestibular input (including the temporoparietal junction, parietal cortex, cingulate gyrus, posterior parietal cortex, hippocampus, and parahippocampal cortex) are altered [10]. These changes in structure and function can result in the upregulation of non-vestibular area networks to support and maintain vestibular perception, leading to cortical reorganization in non-vestibular perception areas, and causing related complications in cognitive and neural function decline, ultimately affecting cognitive function [11]. Patients typically exhibit confusion, delayed reaction, and difficulty with memory and attention [12]. Studies have shown that patients with vestibular-related dizziness or vertigo are four times more likely to have cognitive dysfunction than normal individuals, with 12% of patients experiencing restricted activities due to memory problems or brain fog [13].

Currently, various tools for assessing neurocognitive function are available and can be divided into subjective and objective evaluation tools [14]. Among them, the MMSE scale is one of the most widely used cognitive screening tests in the world and has been promoted and used in multiple countries with various language versions [15]. This test is usually used to evaluate the cognitive function of people aged 60 years and above, and studies have shown that its sensitivity is 85% and specificity is 90% [16]. The MMSE scale is simple, easy to administer, takes a short time, and can evaluate cognitive function from multiple dimensions, including orientation, memory, attention and calculation, recall, and language ability, with a total score of 30 points. However, MMSE scores can be affected by factors such as age, gender, and education level [17]. In addition, the scale is also influenced by population characteristics [18]. In China, according to the education level of the subjects, scores of ≤21 for illiterate and primary school, ≤27 for junior high school, and ≤28 for high school and above are qualitatively considered as cognitive impairment. According to the age of the subjects, scores of ≤27 for those aged 57–69 years, ≤23 for those aged 70–80 years, and ≤21 for those aged 81–97 years are qualitatively considered as cognitive impairment [19].

Event-related potentials (ERPs) are a type of objective cognitive function test in electrophysiology, consisting of endogenous and exogenous components. Different electrophysiological components represent different cognitive stages that individuals undergo while performing the test task [20]. Among them, P300 is an endogenous component of ERPs used to evaluate the neurobiological indicators of the brain’s initial cognitive processing of information. P300 is usually induced under an oddball stimulation pattern [21], in which two distinguishable stimuli are presented in random order, and individuals only need to respond when a rare target stimulus appears. P300 appears about 300 ms after the stimulus onset and includes two neural response components: P3a and P3b. P3a is interpreted as an index of the orienting response of positive attention and can activate the frontal brain area [22], while P3b is interpreted as allocating attentional resources for updating working memory and has a closer relationship with temporoparietal activity [23]. P300 is influenced by attention, discrimination, and memory and reflects the activity of the cerebral cortex, with the latency reflecting the rate of stimulus conduction and the amplitude reflecting the subject’s memory and attention [24]. In the field of neuroscience, P300 has been widely used as an objective electrophysiological test for cognitive function evaluation in diseases such as stroke, brain injury, Alzheimer’s disease, and schizophrenia [25]. For patients with cognitive impairments, a decrease in P300 amplitude, prolongation of latency, or inability to elicit it have become reliable diagnostic indicators of the disease.

Although the literature has reported that vestibular injury can cause dizziness and vertigo, and affect cognitive function [26], few studies have explored the relationship between P300 and vestibular injury-related cognitive function. Therefore, this study aimed to use the Dizziness Handicap Inventory (DHI) to group patients with different degrees of dizziness and compare their cognitive function impairments. At the same time, we also attempted to establish the correlation between P300 objective electrophysiological examination and subjective cognitive function evaluation questionnaires, in order to use P300 for the cognitive function evaluation of vestibular-related dizziness and vertigo patients. To our knowledge, this is the first study to use P300 to investigate the cognitive function of patients with vestibular vertigo diseases. We found that the more severe the dizziness, the more obvious the patient’s anxiety and depression emotions, and the longer the latency and the lower the amplitude of P300. P300 can be used as an objective electrophysiological examination to evaluate the cognitive function of patients with vestibular-related dizziness, and vertigo and is a useful supplement to routine subjective cognitive function evaluation scales.

## 2. Materials and Methods

Seventy-nine elderly patients with vertigo imbalance were selected and categorized into mild, moderate, and severe groups according to the Dizziness Handicap Inventory (DHI). Depression, anxiety, and cognitive functioning of these patients were assessed using the auditory P300 component of event-related potentials (ERPs), Generalized Anxiety Disorder Questionnaire-7 (GAD-7), Patient Health Questionnaire-9 (PHQ-9), and Mini-Mental State Exercise (MMSE), respectively.

### 2.1. Subjects

This study selected a total of 79 elderly patients aged 60 or above diagnosed with vestibular-related dizziness or vertigo at the Vertigo Center of Xinhua Hospital, affiliated with Shanghai Jiao Tong University School of Medicine, from January to December 2022. Among them, there were 35 male patients (44.3%) and 44 female patients (55.7%), with an age range of 60 to 89 years and a mean age of 68.40 ± 5.08 years. Before testing, all patients were informed of the relevant procedures and underwent routine vestibular function and hearing tests, as well as comprehensive otolaryngological and neurological history and physical examinations. Vestibular function tests included eye movement examination, caloric testing, video head impulse testing, subjective visual vertical, and VEMP testing. Hearing tests included impedance audiometry, pure-tone audiometry, auditory brainstem response, and P300 event-related potential. The inclusion criteria were as follows: (1) diagnosis of vestibular-related dizziness or vertigo; (2) patients aged 60 or above with a disease course of more than 3 months; (3) able to communicate normally in Mandarin and able to cooperate actively in completing various tests and questionnaires. The exclusion criteria were as follows: (1) incomplete data; (2) patients previously diagnosed with dementia, mild cognitive impairment, or stroke; (3) patients with a disease course of less than 3 months or aged less than 60 years; (4) patients with severe visual or hearing impairments, unable to communicate normally, cooperate in testing, or complete questionnaires actively; (5) patients who have undergone MRI scanning and been diagnosed with organic brain lesions such as brain tumors or cerebrovascular lesions; (6) patients who have taken psychotropic drugs such as steroids and antidepressants in the last three months.

This study received approval from the Ethics Review Committee of Xinhua Hospital, affiliated to Shanghai Jiao Tong University School of Medicine. This study is a single-center clinical study, and all patients were evaluated by an otolaryngologist who has been engaged in the diagnosis and treatment of inner-ear vestibular-related diseases for more than 10 years.

### 2.2. Methods

#### 2.2.1. Subjective Cognitive Function Evaluation

All patients were assessed using the Chinese version of the MMSE questionnaire [27], which was administered by professional personnel in face-to-face evaluations. The questionnaire includes five different dimensions, which assess orientation (10 points, including time and place), memory (3 points, assessing immediate memory ability), attention and calculation (5 points, assessing calculation accuracy), recall ability (3 points, assessing recall ability for the previous three items), and language ability (9 points, including cognitive function for naming ability, repetition ability, reading ability, three-step command, writing ability, and structural ability). During the assessment, the patient’s age and educational level were taken into account.

#### 2.2.2. Subjective Symptom Assessment

For all patients, the Chinese version of the Dizziness Handicap Inventory (DHI) scale was used to evaluate the symptoms and extent of dizziness or vertigo. The DHI scale includes 3 dimensions: physical, functional, and emotional, with a total of 25 items. The choice for each item is “Yes”, “Sometimes”, or “No”. The corresponding scores are 4, 2, and 0 points, totaling 100 points, which is used to valuate the severity of dizziness [28]. Previous studies have established 3 classifications with different degrees of vertigo according to the score on the DHI scale: 0–30 is mild, 31–60 is moderate, and 61–100 is severe [29]. At the same time, the degree of vertigo is supplemented by the use of a visual analog scale (VAS). Here, 0 points are no dizziness and 10 points are severe dizziness. Patients are scored according to their subjective feelings of vertigo.

We used the Chinese version of GAD-7 to evaluate the anxiety of patients. There are a total of 7 questions. The choices for each question are “no”, “a few days”, “more than half of the time”, and “almost every day”. The corresponding scores are 0, 1, 2, and 3, totaling 21 points. Here, 0–4 points are no anxiety disorder, 5–9 points may be a mild anxiety disorder, 10–13 points may be a moderate anxiety disorder, 14–18 points may be a moderate–severe anxiety disorder, and 19–21 points may be severe anxiety disorder [30].

We used the Chinese version of PHQ-9 to evaluate the depression of patients. There are a total of 9 questions. The choices for each question are “no”, “a few days”, “more than half of the time”, and “almost every day”. The corresponding scores are 0, 1, 2, and 3, totaling 27 points. Here, 0–4 points are no depression, 5–9 points may be mild depression, 10–14 points may be moderate depression, 15–19 points may be moderate–severe depression, and 20–27 points may be severe depression [30].

#### 2.2.3. P300 Mean Latency and Amplitude of Event-Related Potentials

This study used the international standard hearing test equipment Eclipse EAEP module (Denmark) to record the P300 potential. The polar electrode was attached to the subject’s forehead (Fpz), the recording electrode was attached to the vertex (Cz), and the reference electrode was attached to the left and right mastoids (M1 and M2). The interelectrode resistance was controlled to be ≤5 kΩ. During the test, each stimulus window lasted 700 milliseconds, with a high-pass filter cutoff frequency of 1 Hz, a low-pass filter cutoff frequency of 17 Hz, and a stimulus repetition rate of 0.6 s per trial, and more than 100 trials were presented. The ER-3A insert earphones were used to provide auditory stimuli, with a sound intensity of 70 dB nHL.

The subjects were required to stay awake and lie flat on the testing bed throughout the entire testing process. In a series of standard and target stimulus presentations, the subjects were instructed to pay attention to the target stimuli and count them. The frequencies of the standard and target stimuli were 1000 Hz and 2000 Hz, respectively, presented in an oddball paradigm, where the probability of target stimuli was 20% and the probability of non-target stimuli was 80%. The subjects needed to lie supine in a standard soundproof room with a background noise level of <20 dB (A), relax as much as possible, avoid blinking and swallowing, and concentrate their attention. Prior to the test, the subjects were required to listen to a sound consisting of short pure tones of 1 kHz and 2 kHz and were told to memorize the number of target stimuli and ignore the non-target stimuli. These two stimuli were presented randomly at a certain interval. Before the formal test, the subjects were familiarized with the entire testing process through 2–3 pre-tests. Each subject was tested twice, with an interval of 3–5 min between the two tests, and the error between the two tests should not exceed 10%. The P300 amplitude was defined as the peak-to-peak difference between the average baseline voltage and the maximum positive ERP peak before the preceding negative trough between 250 and 500 milliseconds after stimulus onset, in microvolts. In this time window, the P300 component was defined as the earlier and larger peak elicited by deviant stimuli. [23,31].

## 3. Statistical Analyses

We conducted statistical analyses on the data using SPSS 26.0 and created graphs using Prism 9 Version 9.4.1. Descriptive statistics for quantitative data are presented as means ± standard deviations (M ± SD). For independent data that were normally distributed with homogenous variances, we used t-tests to compare means; for multiple independent groups, we used ANOVA with the Bonferroni correction. For non-normally distributed or heteroscedastic data, we used nonparametric tests such as the Wilcoxon rank-sum test for two independent groups and the Kruskal–Wallis test with the Bonferroni correction for multiple independent groups. For non-independent data, we used paired *t*-tests to compare differences between pairs, assuming that the data were normally distributed. We also conducted Spearman’s correlation analyses to evaluate the correlations between P300 amplitude and latency and DHI, MMSE, GAD-7, and PHQ-9. A correlation was considered significant if the correlation coefficient (r) was greater than 0.5 and the *p*-value was less than 0.05.

## 4. Results

The results we obtained are as follows.

### 4.1. Subject Characteristics

According to the inclusion criteria, there were a total of 79 participants in this study, with 35 males (44.3%) and 44 females (55.7%). Based on the Dizziness Handicap Inventory (DHI) scores, the participants were categorized into three groups: mild, moderate, and severe, with 20 cases (25.3%) in the mild group, 40 cases (50.6%) in the moderate group, and 21 cases (26.6%) in the severe group. There were no significant differences in gender distribution among the three groups (*p* > 0.05). Regarding educational background, 3 participants (3.8%) had less than 6 years of education, 7 participants (8.9%) had 6–9 years of education, and 69 participants (87.3%) had more than 9 years of education, with no significant differences among the three groups (*p* > 0.05). In terms of age, 19 participants (24.1%) were between 60 and 70 years old, 28 participants (35.4%) were between 70 and 80 years old, and 32 participants (40.5%) were over 80 years old, with no significant differences in age distribution among the three groups (*p* > 0.05). Please refer to Table 1 for details.

### 4.2. The Comparison of MMSE Scale Scores among Patients with Different Degrees of Impairment

According to the MMSE scale scores, cognitive function was evaluated in three groups of patients (a total of 38 individuals). In the mild group, 3 individuals had cognitive impairment (3/20, 15%), while in the moderate group, 20 individuals had cognitive impairment (20/39, 51.3%), and in the severe group, 12 individuals had cognitive impairment (12/20, 60%). The difference in abnormal rates between the three groups was statistically significant (*p =* 0.005). Further pairwise comparisons revealed that the difference in abnormal rates between the mild group and moderate group was statistically significant (*p =* 0.006), as was that between the mild group and severe group (*p =* 0.009). However, there was no statistically significant difference in abnormal rates between the moderate and severe groups (*p =* 1.000), as shown in Figure 1. The total MMSE scale score difference among the three groups was statistically significant (*p =* 0.016), but there was no statistically significant difference in orientation ability (*p =* 0.141), while memory ability difference was statistically significant (*p =* 0.047), and there were no statistically significant differences in attention and calculation ability (*p =* 0.285), recall ability (*p =* 0.079), or language ability (*p =* 0.200) among the three groups, as shown in Table 2.

### 4.3. The Comparison of GAD-7 and PHQ-9 Scale Scores among Different Groups

The GAD-7 scores for different severity groups were 1.13 ± 0.44, 5.44 ± 0.76, and 5.83 ± 1.08, respectively. There was a statistically significant difference in the scores between these groups (*p =* 0.007). Further pairwise comparisons showed that the difference between the mild group (1.13 ± 0.44) and the moderate group (5.44 ± 0.76) was statistically significant (*p =* 0.010), as was the difference between the mild group (1.13 ± 0.44) and the severe group (5.83 ± 1.08) (*p =* 0.015). However, there was no statistically significant difference between the moderate group (5.44 ± 0.76) and the severe group (5.83 ± 1.08) (*p =* 1.000). See Figure 2A for details.

Similarly, the PHQ-9 scores for different severity groups were 2.50 ± 0.50, 5.92 ± 0.56, and 6.67 ± 0.74, respectively, with a statistically significant difference in scores between these groups (*p =* 0.003). Further pairwise comparisons showed that the difference between the mild group (2.50 ± 0.50) and the moderate group (5.92 ± 0.56) was statistically significant (*p =* 0.006), as was the difference between the mild group (2.50 ± 0.50) and the severe group (6.67 ± 0.74) (*p =* 0.004). However, there was no statistically significant difference between the moderate group (5.92 ± 0.56) and the severe group (6.67 ± 0.74) (*p =* 1.000). See Figure 2B for details.

### 4.4. The Comparison of P300 Latency and Amplitude among Different Groups

The P300 latencies of the different severity groups were 292 ± 10 ms, 301 ± 8 ms, and 328 ± 5 ms, respectively. The differences in P300 latency among the different severity groups were statistically significant (*p =* 0.010). Further pairwise comparisons revealed no statistically significant difference in P300 latency between the mild group (292 ± 10 ms) and moderate group (301 ± 8 ms) (*p =* 1.000), but there were statistically significant differences in P300 latency between the mild group (292 ± 10 ms) and severe group (328 ± 5 ms) (*p =* 0.034), and between the moderate group (301 ± 8 ms) and severe group (328 ± 5 ms) (*p =* 0.024). See Figure 3A for details.

The P300 amplitudes of the different severity groups were 14.4 ± 2.6 μV, 3.9 ± 0.8 μV, and 5.1 ± 1.4 μV, respectively. The differences in P300 amplitude among the different severity groups were statistically significant (*p =* 0.004). Further pairwise comparisons revealed statistically significant differences in P300 amplitude between the mild group (14.4 ± 2.6 μV) and moderate group (3.9 ± 0.8 μV) (*p =* 0.001) and between the mild group (14.4 ± 2.6 μV) and severe group (5.1 ± 1.4 μV) (*p =* 0.010), but no statistically significant difference in P300 amplitude between the moderate group (3.9 ± 0.8 μV) and severe group (5.1 ± 1.4 μV) (*p =* 1.000). See Figure 3A,B for details.

There were correlations between Dizziness Handicap Inventory (DHI) scores and P300 latency and amplitude in different levels of dizziness (*p =* 0.026, R^2^ = 0.095, *p =* 0.010, R^2^ = 0.126); see Figure 4A,B for details. There was no correlation between Visual Analog Scale (VAS) scores and P300 latency and amplitude in different levels of dizziness (*p =* 0.935, *p =* 0.199); see Figure 4C,D for details.

### 4.5. Correlations of MMSE, GAD-7, PHQ-9, and P300 Latency and Amplitudes

We found no significant correlation between MMSE scores and the GAD-7 or PHQ-9 scale (*p =* 0.619, *p =* 0.184), as shown in Figure 5A,B. Additionally, we found no significant correlation between MMSE scores and P300 latency or amplitude (*p* = 0.883, *p* = 0.104), as shown in Figure 5C,D. For the GAD-7 and PHQ-9 scales, we found no significant correlation with P300 latency (*p* = 0.278, *p* = 0.583), as shown in Figure 6. However, we did find a significant correlation between these scales and P300 amplitude (*p* = 0.007, R^2^ = 0.154, *p* = 0.010, R^2^ = 0.141), as shown in Figure 6.

## 5. Discussion

Damage to the vestibular system has been linked to cognitive impairment [12,13,28,32], but there is limited research on the relationship between the severity of dizziness and cognitive impairment. In this study, we classified dizziness patients into three groups—mild, moderate, and severe—based on their scores on the Dizziness Handicap Inventory (DHI) and compared their scores on the Mini-Mental State Examination (MMSE), the Generalized Anxiety Disorder-7 (GAD-7), and the Patient Health Questionnaire-9 (PHQ-9), as well as their event-related potential (ERP) P300 latency and amplitude. We also analyzed the correlation between these variables.

This study found that there were statistically significant differences in the abnormal rates and scores on the MMSE scale for assessing cognitive impairment among the three groups of patients (*p* = 0.005, *p* = 0.016), with the proportion of cognitive impairment patients in the mild, moderate, and severe groups being 15%, 51.3%, and 60%, respectively. The MMSE scores for the three groups were 28.30 ± 1.66, 26.87 ± 2.13, and 25.8 ± 3.75, respectively. As the severity of dizziness increased, the incidence of cognitive impairment gradually increased and the MMSE score gradually decreased. The differences in abnormal rates (*p* = 0.006, *p* = 0.009) and MMSE scores (*p* = 0.031, *p* = 0.032) between the mild group and the moderate and severe groups were statistically significant, indicating that the severity of dizziness affects cognitive impairment. However, there was no statistically significant difference in abnormal rates and scores between the moderate and severe groups (*p* = 1.000, *p* = 1.000), which may be related to the sensitivity of the MMSE scale. Previous studies have shown that the sensitivity of the MMSE scale is not high enough for patients with mild cognitive impairment [33,34]. In comparison, the MoCA scale has higher sensitivity for patients over 60 years old [35]. However, the MMSE scale is widely used in clinical practice due to its short administration time and ease of use [36,37]. Memory impairment is the earliest symptom of MCI, and only the memory dimension of the MMSE scale showed a significant difference in this study (*p* = 0.047), while the other dimensions showed no significant difference, which suggests that most of the cognitive impairment patients included in this study had mild cognitive impairment [38]. Furthermore, there was no significant correlation between MMSE scores and P300 latency or amplitude in this study, which may be due to the low discriminant ability of the MMSE scale and the concentration of scores. Similar phenomena have been observed by Toyoshima et al., who found that scores in subjective assessment were not significantly correlated with P300 amplitude, while objective neurophysiological examinations were significantly correlated [25]. These findings suggest that P300 neurophysiological evaluation plays an important role in studying cognitive impairment and should be a useful complement to routine clinical assessment.

It is reported that vestibular disorders such as vestibular migraine, vestibular paroxysmia, and Meniere’s disease may cause anxiety and depression symptoms (49). Yuan et al.’s research suggests that patients with vestibular migraine and Meniere’s disease have a higher risk of anxiety and depression than those with benign paroxysmal positional vertigo and vestibular neuritis [39]. However, patients with chronic unilateral and bilateral vestibular lesions may not experience psychological disorders such as anxiety and depression [40]. Li et al. studied 559 patients with dizziness and found that the longer the duration of dizziness, the higher the incidence of anxiety and depression, and the more likely it is for anxiety to occur. The results of this study show statistically significant differences in the scores on the GAD-7 and PHQ-9 scales for varying degrees of dizziness (*p* = 0.007, *p* = 0.003). The GAD-7 scores for the mild, moderate, and severe groups were 1.13 ± 0.44, 5.44 ± 0.76, and 5.83 ± 1.08, respectively. The PHQ-9 scores for the mild, moderate, and severe groups were 2.50 ± 0.50, 5.92 ± 0.56, and 6.67 ± 0.74, respectively. As the severity of dizziness increased, so did the severity of anxiety and depression. Scores of 0–4 indicate no anxiety or depression, while scores of 5–9 indicate possible mild anxiety or depression. Therefore, mild dizziness patients may not experience anxiety or depression, while those with moderate to severe dizziness may have mild anxiety or depression. Anxiety and depression have been linked to dizziness and cognitive dysfunction [13,41] and may mediate and worsen the impact of dizziness on cognitive dysfunction [42]. As expected, our results indicate that patients with higher levels of psychological distress reported a higher incidence of cognitive dysfunction, which is consistent with previous research findings [12,42]. We did not find a correlation between the GAD-7 and PHQ-9 scores and MMSE scores (*p* = 0.619, *p* = 0.184), but we did find a significant correlation between the GAD-7 and PHQ-9 scores and P300 amplitude (*p* = 0.007, *p* = 0.010). As anxiety and depression increased, the P300 amplitude gradually decreased. We speculate that the reason for this result may be that a single questionnaire survey may not comprehensively evaluate a patient’s mental and cognitive status. Additionally, our study once again illustrates that subjective cognitive function assessment questionnaires may not fully reflect a patient’s cognitive function, especially in the early stages of cognitive dysfunction. In contrast, P300 evoked potential based on neurophysiology may be better suited for early monitoring and assessing the reorganization of central cognitive-related neural networks in non-vestibular cortical areas.

P300 is a low-cost, non-invasive, and reproducible test of psychological–neurophysiological response with high sensitivity. It can be used to monitor the neuro-electrophysiological activity of the brain in response to specific sensory, cognitive, or motor events from the external environment, and is not affected by culture or education level [43]. P300 is an endogenous component of ERPs that reflects the initial cognitive processing of information in the brain [21]. Its latency and amplitude changes are considered the most valuable electrophysiological objective indicators for evaluating cognitive function. Latency and amplitude can reflect the speed and attention of brain information processing, which are crucial for cognitive function. P300 latency is considered a standard for measuring stimulus classification speed [44,45] and is related to individual differences and psychological function speed [46]. P300 amplitude depends on the formation of synchronous discharge of a large number of neurons and is used to guide resource allocation of attention [31]. Zeng et al. studied 83 patients with idiopathic inflammatory–demyelinating diseases, including 24 with multiple sclerosis (MS), 37 with neuromyelitis spectrum disorders (NMOSD), 11 with clinically isolated syndrome (CIS), and 11 with acute disseminated encephalomyelitis (ADEM), and found that the P300 latency was prolonged and the amplitude was decreased in patients with cognitive impairment [47]. Khedr et al. studied 20 patients with systemic sclerosis and found that the P300 latency was prolonged in patients with cognitive impairment [48]. Li et al. found that the P300 amplitude of 227 traumatic brain injury patients with cognitive impairment was lower than that of normal individuals [49]. All of the above studies indicate that the P300 latency is prolonged and the amplitude is decreased in patients with cognitive impairment. However, it should be noted that EPRs are not a specific method and that they can also show abnormalities in lesions that are not initially cognitively impaired or even in diseases that do not initially involve the nervous system [50,51,52,53]. This study found that the differences in P300 latency and amplitude among patients with different degrees of dizziness were statistically significant (*p* = 0.010, *p* = 0.004). As the degree of dizziness increased, the P300 latency was gradually prolonged (*p* = 0.026, R^2^ = 0.095) and the amplitude decreased (*p* = 0.010, R^2^ = 0.126). This indicates that the higher the degree of dizziness, the higher the likelihood of cognitive impairment.

## 6. Study Limitations

This study found that as the severity of dizziness increases, the incidence of cognitive impairment increases, and objective electrophysiological indicators such as prolonged P300 latency and reduced amplitude have important clinical value. However, this study has several limitations: firstly, the group of vestibular-related dizziness or vertigo diseases includes various types of diseases, and different types of vestibular disorders’ impact on cognitive function was not differentiated. Secondly, not all patients underwent P300 testing, which may result in biased results. Thirdly, the MMSE scale used in this study is less sensitive than MoCA for the diagnosis of MCI; thus, in the future, a variety of cognitive assessment scales should be combined to comprehensively evaluate the cognitive function status of patients with different degrees of dizziness and the correlation with P300 event-related potentials. Fourthly, the differences in P300 latency between the mild and moderate groups and the differences in amplitude between the moderate and severe groups in this study were not statistically significant; therefore, a larger sample size and more sensitive dizziness severity grading tools are needed to group patients with different degrees of dizziness to better reveal the differences in P300 latency and amplitude. Finally, the latency and amplitude of P300 are mainly used in the published literature for the assessment of cognitive function using P300, so we also just used the latency and amplitude of P300 as the final clinical evaluation index and conducted the relevant statistical analyses based on the previous reports. However, other metrics of P300 may also provide some important information to evaluate the cognitive function, and it is necessary to determine more details and justifications about the metrics of P300.

## 7. Conclusions

Vestibular-related dizziness or vertigo can affect the cognitive function of patients, and the more severe the vertigo, the higher the possibility of cognitive dysfunction. To objectively evaluate the cognitive function of patients with vestibular-related dizziness or vertigo, including attention, short-term memory, and information processing speed, P300 testing can be used as an effective electrophysiological method. Vestibular-related cognitive damage is associated with prolonged latency and decreased amplitude of P300. P300 testing is of great significance for evaluating the cognitive function of patients with vestibular-related dizziness or vertigo.

## Figures and Tables

**Figure 1 biomedicines-11-02365-f001:**
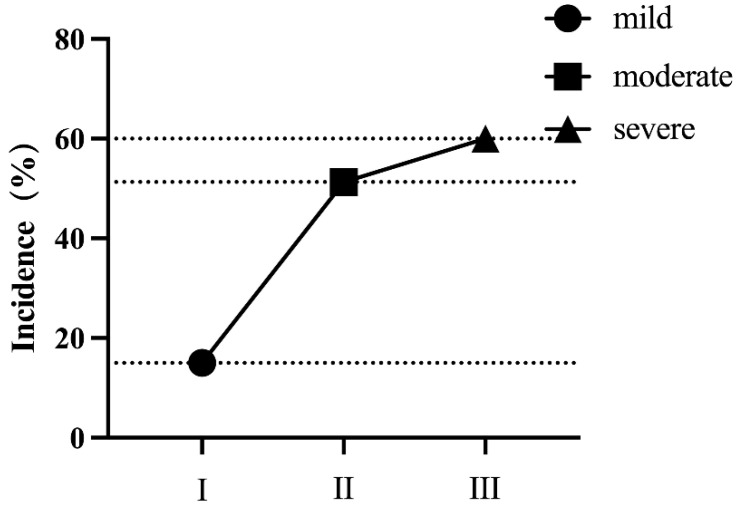
Comparison of abnormal incidence in the MMSE scale among three groups of patients with varying degrees of dizziness. There were statistically significant differences between the mild group and the moderate/severe groups, while no statistical difference was observed between the moderate and severe dizziness groups.

**Figure 2 biomedicines-11-02365-f002:**
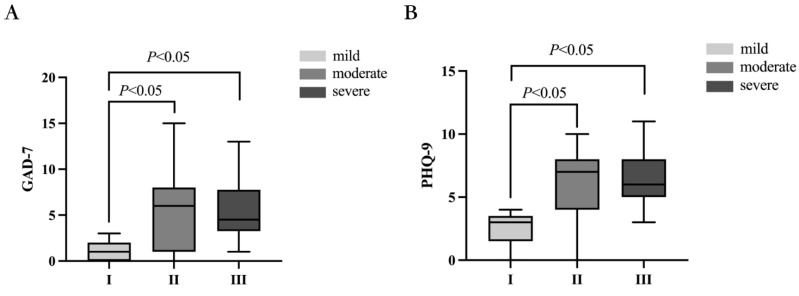
Comparison of GAD-7 and PHQ-9 scale scores among patient groups with varying degrees of impairment. (**A**) There was a statistically significant difference in GAD-7 scores among the three groups. (**B**) There was also a statistically significant difference in PHQ-9 scores among the three groups. *p* < 0.05 indicates a statistically significant difference.

**Figure 3 biomedicines-11-02365-f003:**
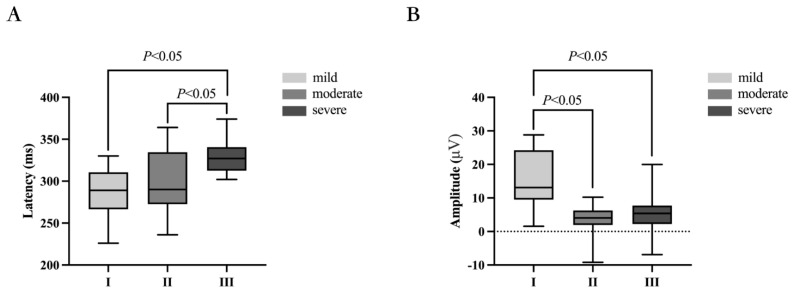
Comparison of P300 latency and amplitude in different degree groups. (**A**) Comparison of the P300 latencies of different groups. The silver box plot represents group I (n = 10), nickel box plot represents group II (n = 26), and iron box plot represents group III (n = 16). Significant differences were observed between group I and group III and between group II and group III. There was no significant difference between group I and group II. (**B**) Comparison of the P300 amplitude of different groups. The silver box plot represents group I (n = 10), nickel box plot represents group II (n = 26), and iron box plot represents group III (n = 16). Significant differences were observed between group I and group II and between group I and group III. There was no significant difference between group II and group III. Group I: mild degree in DHI; group II: moderate degree in DHI; group III: severe degree in DHI.

**Figure 4 biomedicines-11-02365-f004:**
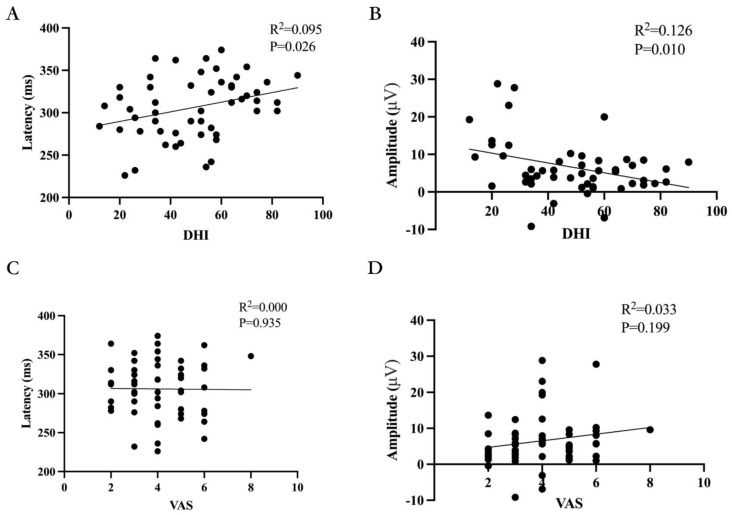
Correlations of DHI and VAS with P300 latency and amplitudes. (**A**) The relationship between severity of Dizziness Handicap Inventory (DHI) and P300 latency in ERPs. Spearman’s correlation was performed and revealed that DHI had a significant positive correlation with P300 latency. (**B**) The relationship between severity of Dizziness Handicap Inventory (DHI) and P300 amplitude in ERPs. Spearman’s correlation was performed and revealed that DHI had a significant negative correlation with P300 amplitude. (**C**) The relationship between Visual Analogue Scale (VAS) and P300 latency in ERPs. Spearman’s correlation was performed and revealed that VAS had no significant correlation with P300 latency. (**D**) The relationship between Visual Analogue Scale (VAS) and P300 amplitude in ERPs. Spearman’s correlation was performed and revealed that VAS had no significant correlation with P300 amplitude.

**Figure 5 biomedicines-11-02365-f005:**
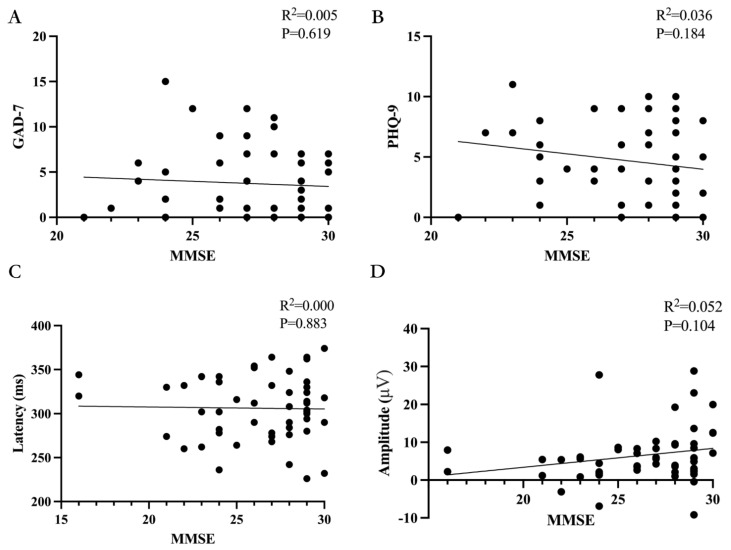
Correlations of MMSE, GAD-7/PHQ-9, and P300 latency and amplitudes. (**A**) The relationship between MMSE and GAD-7. Spearman’s correlation was performed and revealed that MMSE had no significant correlation with GAD-7. (**B**) The relationship between MMSE and PHQ-9. Spearman’s correlation was performed and revealed that MMSE had no significant correlation with PHQ-9. (**C**) The relationship between MMSE and P300 latency in ERPs. Spearman’s correlation was performed and revealed that MMSE had no significant correlation with P300 latency. (**D**) The relationship between MMSE and P300 amplitude in ERPs. Spearman’s correlation was performed and revealed that MMSE had no significant correlation with P300 amplitude.

**Figure 6 biomedicines-11-02365-f006:**
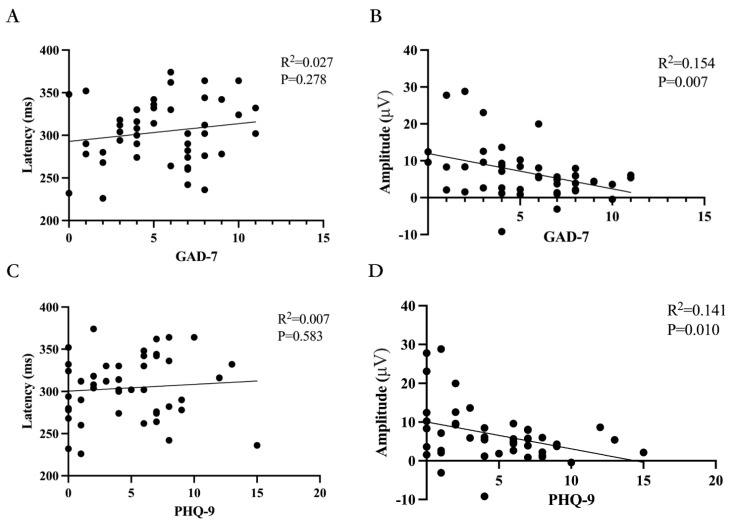
Correlations of GAD-7 and PHQ-9 with P300 latency and amplitudes. (**A**) The relationship between GAD-7 and P300 latency in ERPs. Spearman’s correlation was performed and revealed that GAD-7 had no significant correlation with P300 latency. (**B**) The relationship between GAD-7 and P300 amplitude in ERPs. Spearman’s correlation was performed and revealed that GAD-7 had a significant positive correlation with P300 amplitude. (**C**) The relationship between PHQ-9 and P300 latency in ERPs. Spearman’s correlation was performed and revealed that PHQ-9 had no significant correlation with P300 latency. (**D**) The relationship between PHQ-9 and P300 amplitude in ERPs. Spearman’s correlation was performed and revealed that PHQ-9 had a significant positive correlation with P300 amplitude.

**Table 1 biomedicines-11-02365-t001:** Basic information on patients in different severity groups.

Variables		I	II	III	*p*-Value
Gender	Male	7	20	8	0.445
	Female	13	19	12	
Education	<6 Y	1	1	1	0.408
	6–9 Y	4	2	1	
	>9 Y	15	36	18	
Age group	60–70 Y	7	8	4	0.798
	70–80 Y	7	15	6	
	>80 Y	6	16	10	

Note: I: mild degree group; II: moderate degree group; III: severe degree group. Y: year.

**Table 2 biomedicines-11-02365-t002:** Different dimensions of MMSE scales of different severity groups ($\bar{x}\pm s$).

	I	II	III	*p*-Value
Orientation	9.80 ± 0.41	9.38 ± 0.81	9.45 ± 1.10	0.141
Registration	2.95 ± 0.22	3 ± 0 ^a^	2.85 ± 0.37 ^a^	0.047
Attention and Calculation	4.65 ± 0.75	4.28 ± 0.97	4.15 ± 1.39	0.285
Recall	2.85 ± 0.37	2.49 ± 0.79	2.30 ± 0.92	0.079
Language	8.05 ± 0.83	7.72 ± 0.92	7.05 ± 1.85	0.200
MMSE	28.30 ± 1.66 ^bc^	26.87 ± 2.13 ^b^	25.8 ± 3.75 ^c^	0.016

Note: I: mild degree group; II: moderate degree group; III: severe degree group. a: Registration comparison with II and III. b: MMSE comparison with I and II. c: MMSE comparison with I and III. Pa = 0.040, Z = 2.472, Pb = 0.031, Z = 2.560, Pc = 0.032, Z = 2.548.

## Data Availability

The original contributions presented in this study are included in the article. Further inquiries can be directed to the corresponding authors.

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
