# Peer review of "P300 Event-Related Potential Predicts Cognitive Dysfunction in Patients with Vestibular Disorders"

_biomedicines, 2023, doi:10.3390/biomedicines11092365_

Round 1

Reviewer 1 Report

The paper presented to me for review addresses the interesting issue of P300 event-related potential predicts cognitive dysfunction in patients with vestibular disorders. The application of ERP is still small despite the simplicity of this method.

The paper is written in good language, the different parts of the manuscript are typical, logical and understandable to the reader.

However, before accepting the paper for publication, I have some comments that should be included in the revision:

1. the patients did not have a brain imaging study (unless they did and the authors do not write about it), which is a significant limitation of the paper because abnormalities in either ERP or neuropsychological testing may be due, for example, to vascular or other organic brain changes

2. did the patients included in the study take any medications that could potentially affect the bioelectrical function of the brain? e.q. steroids, neuroleptics, antidepressants, antipsychotics or others? this should be an exclusion criterion

3. in the discussion, it should be emphasized that EPRs are not a specific method and can also show abnormalities in diseases that do not have an initial course of cognitive impairment and even do not involve the nervous system initially, for example, in endocrine or autoimmune diseases based on: PMID: 33510336, PMID: 26271272, PMID: 37255838 and PMID: 35820509.

Author Response

Response to Reviewer 1 Comments

Dear reviewer,

Thank you for your careful consideration of our manuscript and we appreciate you for all the comments. After carefully reviewing the comments, we have revised the manuscript and responded to all the comments. All amendments are indicated in red characters in the revised manuscript. Our point-by-point responses to the comments are attached below.

Point 1: The patients did not have a brain imaging study (unless they did and the authors do not write about it), which is a significant limitation of the paper because abnormalities in either ERP or neuropsychological testing may be due, for example, to vascular or other organic brain changes.

Response 1: Thank you for your insightful comment. All patients who were enrolled in this study underwent an MRI scanning of the brain to rule out brain-related lesions. we have added this important information in the manuscript, please see Lines 146-148.

Point 2: Did the patients included in the study take any medications that could potentially affect the bioelectrical function of the brain? e.q. steroids, neuroleptics, antidepressants, antipsychotics or others? this should be an exclusion criterion

Response 2: Thank you for the insightful comment. We totally agree with you that medications could potentially affect the bioelectrical function of the brain. All patients enrolled in this study did not take any steroids, neuroleptics, antidepressants, or antipsychotic medications during the last three months. We have added several sentences to clearly stated these exclusion criteria according to your suggestion, please see Lines 148-149.

Point 3:  In the discussion, it should be emphasized that EPRs are not a specific method and can also show abnormalities in diseases that do not have an initial course of cognitive impairment and even do not involve the nervous system initially, for example, in endocrine or autoimmune diseases based on: PMID: 33510336, PMID: 26271272, PMID: 37255838 and PMID: 35820509.

Response 3: Thank you for your valuable suggestion. We have added this sentence in the revised version. Please see Lines 455-458.

Reviewer 2 Report

This paper describes a details analysis of the relation between P300 information and cognitive dysfunction. The paper does not propose any new technique but the analyses are interesting and the paper is well presented.

Comments to improve the paper:

·       At the end of the introduction, I’d suggest including a list with the main contributions.

·       In any section, before a subsection title, there must be an introductory paragraph. For examples, Section 2 and section 2.1.

·       You have considered the P300 latency and amplitude. Are there other metrics that could be analysed? More details and justifications about these metrics are need.

·       I think it would be interesting also consider participants without any problem (control group).

·       I miss a more details analysis between vestibular disorder and cognition dysfunction.

Author Response

Response to Reviewer 2 Comments

Dear reviewer,

Thank you for your careful consideration of our manuscript and we appreciate you for all the comments. After carefully reviewing the comments, we have revised the manuscript and responded to all the comments. All amendments are indicated in red characters in the revised manuscript. Our point-by-point responses to the comments are attached below.

Point 1: At the end of the introduction, I’d suggest including a list with the main contributions.

Response 1: Thank you for your valuable suggestion, and we have added the main contributions based on this study to clinical practice at the end of the introduction in the revised version. Please see lines 115-120.

Point 2: In any section, before a subsection title, there must be an introductory paragraph. For examples, Section 2 and section 2.1.

Response 2: Thank you for your comments, we have added the introductory paragraph between section 2 and section 2.1, and between 4 and 4.1. please see lines 122-127 and 233.

Point 3:  You have considered the P300 latency and amplitude. Are there other metrics that could be analysed? More details and justifications about these metrics are need.

Response 3: Thank you for your insightful comments. In the published literature on the assessment of cognitive function by P300, the latency and amplitude of P300 are mainly used, so in the literature, we also just used the latency and amplitude of the P300 as the final clinical evaluation index and did the relevant statistical analyses based on the previous reports. Just like your suggestion, other metrics of P300 may also provide some important information to evaluate the cognitive function, and it is needed to determine more details and justifications about the metrics of P300. In future studies, we will follow your suggestion to further collect some other biological indexes of P300 for analysis, in order to provide more valuable clinical information on the assessment of cognitive function by P300. We also state these limitations in the discussion of study limitations. Please see lines 478-484.

Point 4: I think it would be interesting also consider participants without any problem (control group).

Response 4: Thank you for your comments. In this study, we just want to determine the degree of vertigo on cognitive function, so we only compared the differences between older adults with different levels of vertigo. In future studies, we will further include the normal group for comparative studies according to your suggestions.

Point 5: I miss a more details analysis between vestibular disorder and cognition dysfunction.

Response 5: We found that the more severe the dizziness, the more obvious the patient's anxiety and depression emotions, and the more severe the dizziness, the more serious the cognitive impairment, so there is a close correlation between dizziness and cognitive impairment. We also found that the severity of dizziness is related to various indicators of P300. For example, the more severe the dizziness, the longer the latency and the lower the amplitude of P300. P300 is a recognized objective electrophysiological examination to evaluate the cognitive function of patients, so P300 can also be used to assess or predict cognitive impairment with vestibular-related dizziness and vertigo and is a useful supplement to routine subjective cognitive function evaluation scales.

Reviewer 3 Report

The Authors of the manuscript investigated the links between cognitive impairment and P300 event-related potential (ERP) in older adults with dizziness and balance disorders. The well-documented results of 79 older adult patients with dizziness and balance disorders showed, among other things, that both P300 latency and P300 amplitude were statistically significantly different between patient groups. The findings, which were linked to data obtained from the Generalized Anxiety Disorder Questionnaire-7 (GAD-7), Patient Health Questionnaire-9 (PHQ-9) and Mini-Mental State Examination (MMSE), led to the conclusion that older patients with more severe symptoms of dizziness and balance disorders are more likely to develop abnormal cognitive functions. Therefore, the P300 can be used as an objective neurophysiological test to assess cognitive function in older patients with dizziness and balance disorders.

The preliminary studies conducted are interesting and worth publishing. They can be used in clinical practice, but one should keep in mind the limitations indicated by the Authors in section 6 (Limitations of the study). For this reason, it would be advisable to continue the study and confirm the results obtained on a larger group of patients. The Authors should also answer several questions:

line 225 - whether there are literature data confirming that correlation coefficients greater than 0.5 can be considered significant,

lines 327-328 (Figure 4 B) - in my opinion, DHI has a significant negative relationship with P300 amplitude,

lines 574-576 (reference 35) - no information on where this publication was published.

Author Response

Response to Reviewer 3 Comments

Dear reviewer,

Thank you for your careful consideration of our manuscript and we appreciate you for all the comments. After carefully reviewing the comments, we have revised the manuscript and responded to all the comments. All amendments are indicated in red characters in the revised manuscript. Our point-by-point responses to the comments are attached below.

Point 1: line 225 - whether there are literature data confirming that correlation coefficients greater than 0.5 can be considered significant

Response 1: This Author confirmed that correlation coefficients greater than 0.5 can be considered significant strongest correlation in DOI:10.1002/alr.22744 and 10.3390/ijerph19053106.

Point 2: lines 327-328 (Figure 4 B) - in my opinion, DHI has a significant negative relationship with P300 amplitude,

Response 2: Thank you for your insight finding. There must be a mistake. We have modified it in the manuscript, see line 333.

Point 3:  lines 574-576 (reference 35) - no information on where this publication was published.

Response 3: Thank you for your comments. We have modified this reference in the manuscript:

Ciesielska, Natalia, Remigiusz Sokołowski, Ewelina Mazur, Marta Podhorecka, Anna Polak-Szabela, and Kornelia Kędziora-Kornatowska. "Is the Montreal Cognitive Assessment (MoCA) test better suited than the Mini-Mental State Examination (MMSE) in mild cognitive impairment (MCI) detection among people aged over 60? Meta-analysis". Psychiatria Polska 50 no. 5 (2016): 1039-1052. doi:10.12740/PP/45368.

Round 2

Reviewer 1 Report

The authors fully addressed my comments and incorporated them into the  manuscript. 

Reviewer 2 Report

The authors have addressed my comments